# COVID-19 Vaccine Refusal and Delay among Adults in Italy: Evidence from the OBVIOUS Project, a National Survey in Italy

**DOI:** 10.3390/vaccines11040839

**Published:** 2023-04-13

**Authors:** Davide Gori, Angelo Capodici, Giusy La Fauci, Marco Montalti, Aurelia Salussolia, Giorgia Soldà, Zeno Di Valerio, Francesca Scognamiglio, Maria Pia Fantini, Julie Leask, Heidi J. Larson, Stefania Profeti, Federico Toth, Jacopo Lenzi

**Affiliations:** 1Department of Biomedical and Neuromotor Sciences, University of Bologna, 40126 Bologna, Italy; 2Susan Wakil School of Nursing and Midwifery, Faculty of Medicine and Health, University of Sydney, Sydney, NSW 2050, Australia; 3Sydney Infectious Diseases Institute, Westmead, NSW 2145, Australia; 4Institute for Health Metrics & Evaluation (IHME), University of Washington, Seattle, WA 98105, USA; 5London School of Hygiene and Tropical Medicine (LSHTM), London WC1E 7HT, UK; 6Department of Political and Social Sciences, University of Bologna, 40125 Bologna, Italy

**Keywords:** COVID-19 vaccine, vaccine uptake, vaccine hesitancy, Italy, policy tools

## Abstract

Background: Vaccine hesitancy was defined by the World Health Organization (WHO) in 2019 as a major threat to global health. In Italy, reluctance to receive vaccines is a widespread phenomenon that was amplified during the COVID-19 pandemic by fear and mistrust in government. This study aims to depict different profiles and characteristics of people reluctant to vaccinate, focusing on the drivers of those who are in favor of and those who are opposed to receiving the COVID-19 vaccine. Methods: A sample of 10,000 Italian residents was collected. A survey on COVID-19 vaccination behavior and possible determinants of vaccine uptake, delay, and refusal was administered to participants through a computer-assisted web interviewing method. Results: In our sample, 83.2% stated that they were vaccinated as soon as possible (“vaccinators”), 8.0% delayed vaccination (“delayers”), and 6.7% refused to be vaccinated (“no-vaccinators”). In general, the results show that being female, aged between 25 and 64, with an education level less than a high school diploma or above a master’s degree, and coming from a rural area were characteristics significantly associated with delaying or refusing COVID-19 vaccination. In addition, it was found that having minimal trust in science and/or government (i.e., 1 or 2 points on a scale from 1 to 10), using alternative medicine as the main source of treatment, and intention to vote for certain parties were characteristics associated with profiles of “delayers” or “no-vaccinators”. Finally, the main reported motivation for delaying or not accepting vaccination was fear of vaccine side effects (55.0% among delayers, 55.6% among no-vaccinators). Conclusion: In this study, three main profiles of those who chose to be vaccinated are described. Since those who are in favor of vaccines and those who are not usually cluster in similar sociodemographic categories, we argue that findings from this study might be useful to policy makers when shaping vaccine strategies and choosing policy instruments.

## 1. Introduction

The rich scientific literature on vaccine hesitancy and its multiple determinants dates back to the pre-COVID-19 era. The World Health Organization (WHO) listed vaccine hesitancy among the ten threats to global health of 2019 and re-defined it as a motivational state of being conflicted or opposed to vaccination [1,2]. Many researchers agree that vaccine hesitancy should not be thought of as a dichotomous variable (yes/no), but rather as a continuum [3,4,5,6,7,8] related both to the individual and the socio-political context in which he/she lives [9,10].

In order to highlight the different beliefs, motivations, and attitudes toward vaccines, some scholars have identified multiple (ideal–typical) “profiles”. In most cases, such psychological profiles have been suggested to define different types of parents with respect to childhood vaccinations. However, such profiles can also be used for adult vaccinations. Expressions, such as “immunization advocate” [11], “vaccine believer” [12], and “unquestioning acceptor” [13] have been used to define the profile of the most pro-vaccination individuals. On the other hand, labels, such as “unconvinced” [12], “rejecter” [14], or “refuser”, [13] have been proposed to identify individuals who are firmly opposed to vaccination. The profiles of the “worried” [11], the “cautious” [12], or the “hesitant” [13] correspond to those who—to varying degrees—are concerned about the adverse effects of vaccination. Some authors have identified as “late vaccinators” [13,14] or “fence-sitters” [4,11,15] those who intentionally postpone vaccination in order to gather more information and assess the effects of the vaccine on other individuals. Other authors have identified “calculator” or “free-rider” profiles [4,16] to classify those who decide whether or not to vaccinate on the basis of a rational calculation between individual costs and benefits.

Investigating ambivalence is inherently complex. Alas, to better understand this phenomenon, it is useful to juxtapose those who have not yet made a decision with those who have. Since data on procrastinators could potentially shed light on vaccine hesitancy as a whole, this study presents data on both those who are in favor of and those who are opposed to receiving the COVID-19 vaccine and expands the subject by analyzing those who are delaying and could therefore be defined as “hesitant”.

Hence, the aim of the present research was to create a specific focus on the different profiles and characteristics of vaccine-reluctant people by using a structured survey conducted on a large national sample of Italy’s demographics.

## 2. Materials and Methods

### 2.1. Study Design and Data Collection

This study was conducted as a cross-sectional computer-assisted web interviewing (CAWI) questionnaire. From 11 April to 29 May 2022, the professional online provider Dynata [17] surveyed 10,000 Italian citizens aged ≥18 years using a stratified sampling based on proportionate allocation by first-level NUTS (Nomenclature of Territorial Units for Statistics) statistical region of residence (Northwest, Northeast, Center, South, and Islands), gender, and age group.

The survey was designed to be completed in ~10 min and included 7 sections investigating demographics and living conditions, data on vaccination against influenza (the focus of this work), pneumococcus, varicella-zoster virus, rotavirus and human papillomavirus, political orientation, and attitudes toward SARS-CoV-2 vaccination, science, and alternative medicine

Data management of Dynata was performed in accordance with the General Data Protection Regulation of the European Union. The survey followed all requirements under Italian regulations.

### 2.2. Variables

Cognitive testing was conducted prior to full implementation, and feedback was used to revise the questionnaire. A total of 15 sociodemographic questions were included: gender; date of birth; region of residence; educational qualification; occupation; living arrangements; ability to pay necessary living expenses; problems with activities of daily living due to physical or mental disabilities; weight; height; suffering from chronic respiratory/cardiovascular disease/diabetes; place where most vaccinations were carried out; preferred place for vaccination; and friends’ and family members’ opinions on vaccination.

Respondents were then asked to indicate which of the following statements they most agreed with: (1) the government should make vaccines available and limit itself to providing information, without trying to influence individual choice; (2) the government should provide incentives for those who vaccinate, but not impose any obligations; (3) the government should provide penalties for those who decide not to vaccinate; and (4) in health emergencies, it is right for the government to mandate vaccination.

Questions on voting intention, trust in science (on an ordinal scale from 1 to 10), trust in government (on an ordinal scale from 1 to 10), behavior towards COVID-19 vaccination, reasons for delaying or refusing vaccination, and use of alternative medicines as the main source of treatment were also included.

The vaccine-specific sections were developed following the domains of the WHO BesD framework [1]: thoughts and feelings, social processes, motivation, practical issues, and vaccination. The questionnaire can be found in the Appendix B.

The data management was performed in accordance with the General Data Protection Regulation of the European Union. The survey experiment followed all requirements under Italian regulations.

### 2.3. Data

The survey was designed to record respondents’ behavior regarding COVID-19 vaccination on the one hand, and to obtain information on the possible determinants of vaccine delay and refusal on the other. Respondents to the questionnaire were asked to indicate which of the following four options represented their situation: (1) I got the vaccine immediately, as soon as it was my turn; (2) I voluntarily postponed vaccination; (3) I did not get the vaccine; (4) I am exempt from vaccination for medical reasons (official medical certification). Based on the answers to this question, it was then decided to divide the sample into three categories: “vaccinators”, “delayers”, and “non-vaccinators”. Study participants were also asked to indicate the main reasons (three options ranked by importance) behind their decisions about vaccination, drawing up the various options on the basis of the “5C Model” of the psychological drivers of vaccine hesitancy (confidence, complacency, constraints, [risk] calculation, and collective responsibility) [7].

Still concerning the determinants of vaccine uptake, delay, and refusal, questions were included to investigate both the main variables established in the literature (sociodemographic conditions, trust in institutions and science, use of alternative medicines) and variables concerning the political domain (politics and policy). Moreover, knowing that perception and utilization of health services may change according to urbanization [18,19], municipalities of residence were subdivided into rural areas (sparsely populated areas), towns or suburbs (intermediate density areas), and cities (densely populated areas) using the Eurostat Degree of Urbanization (DEGURBA) classification system based on 2011 population grids and 2018 administrative boundaries.

### 2.4. Statistical Analysis

All data were summarized as counts and percentages and visualized with the aid of bar charts, pie charts, and thematic maps. To account for under-represented groups in the population due to non-response to the survey, all counts and percentages were weighted using post-stratification [20]. Post-stratification involves adjusting the sampling weights so that they sum to the population sizes within each post-stratum. In practice, we obtained the distribution of the resident adult population of Italy by first-level NUTS statistical region, gender, and age group as of 1 January 2022 (see Appendix A) and post-stratified sampling weights in order to obtain a sample distribution fully in line with the target population (see Appendix A). Unweighted results (that is, not adjusted via post-stratification) are presented in Appendix A for thoroughness.

Differences in sociodemographic characteristics, opinion on vaccination policies, voting intention, and use of alternative medicine according to COVID-19 vaccine uptake status were investigated using the Pearson *χ*^2^ test corrected for the survey design with a second-order correction developed by Rao and Scott (1984) [21,22]. Adjusted residuals, that is, Pearson residuals divided by an estimate of their standard error, were analyzed to assess each cell’s contribution to the overall *χ*^2^ statistic. In particular, residuals beyond ±3.29 (±z_0.001/2_) were considered as significantly implicated in the departure from the null hypothesis of independence between variables.

The Kruskal–Wallis test was used to assess whether trust in the Italian government and science was distributed differently in persons accepting, delaying, and refusing COVID-19 vaccination. Post hoc evaluations were performed with the Conover–Iman test for stochastic dominance among multiple pairwise comparisons, using the Bonferroni–Holm method to control the family-wise error rate [23,24]. As a measure of effect size for the Conover–Iman test, we reported the Harrell’s C index (also known as “common language” effect size), which expresses the average probability that an individual from one of the three groups being compared is more trustful than an individual from one of the other two groups [25].

Respondents medically exempt from being vaccinated were excluded from trust and voting intention analyses due to small numbers. All analyses were carried out using the Stata software package, version 17 [26]. The significance level was set at 0.05, and all tests were two-sided.

## 3. Results

Of the 10,000 respondents to the survey, 8315 (83.2%) claimed to have been vaccinated against COVID-19 as soon as possible and 209 (2.1%) to be exempt from the vaccine, while 803 (8.0%) were “delayers”, i.e., delayed vaccination, and 673 (6.7%) were “no-vaccinators”, i.e., refused to get vaccinated (Figure 1). In the following subsections, results are presented by looking at how “vaccinators”, “delayers”, and “no-vaccinators” were distributed according to each of the potential determinants taken into consideration in the study.

### 3.1. Sociodemographic Characteristics

As shown in Table 1, COVID-19 vaccine uptake was significantly different in men and women (Rao and Scott-corrected *χ*^2^ test = 6.06, *p*-value < 0.001). In particular, as confirmed by post hoc adjusted residuals, vaccine refusal was significantly more common among females than among males (7.5% vs. 5.8%, respectively). With regard to age (Table 1), vaccine acceptance followed a U-shaped distribution (Rao and Scott-corrected *χ*^2^ test = 10.41, *p*-value < 0.001) in which “delayers” were significantly over-represented in the age groups 25–34 (12.4%) and 35–44 years (11.2%) and under-represented above 64 years (4.3%), while “no-vaccinators” were significantly under-represented below 25 years (3.8%).

Educational attainment was also found to be significantly associated with vaccine uptake (Rao and Scott-corrected *χ*^2^ test = 29.65, *p*-value < 0.001) (Table 1). In particular, as confirmed by post hoc adjusted residuals, people refusing vaccination were over-represented among those with less than a high school diploma (9.3%) and under-represented in those with an academic degree (4.7%); in contrast, respondents with education beyond a master’s degree were significantly more inclined to delay vaccination as compared with those with lower education (11.4%) (Table 1).

Lastly, degree of urbanization was found to be significantly associated with vaccine uptake (Rao and Scott-corrected *χ*^2^ test = 18.48, *p*-value < 0.001) (Table 1). In particular, as confirmed by post hoc adjusted residuals, rural areas had a significantly higher (11.3%) percentage of people delaying vaccination than cities and towns or suburbs (7.4% and 7.6%, respectively).

### 3.2. Trust in Science

As illustrated in Figure 2, trust in science followed a significantly different distribution according to vaccine uptake status (Kruskal–Wallis test = 1115.77, *p*-value < 0.001), with significant differences resulting from all post hoc pairwise comparisons across groups (uptake vs. delay: Conover–Iman test = 19.73, *p*-value < 0.001; uptake vs. refusal: Conover–Iman test = 31.10, *p*-value < 0.001; delay vs. refusal: Conover–Iman test = 10.68, *p*-value < 0.001). Indeed, the respondents who exhibited minimal trust (i.e., 1 or 2 points on a scale from 1 to 10) accounted for 18.6% of “no-vaccinators”, 4.7% of “delayers”, and 1.1% of “vaccinators”.

More generally, we estimated a probability of 0.69 (95% CI 0.67–0.71) that a vaccinated individual has a higher trust in science as compared with an individual who puts off getting vaccinated, which increases to 0.83 (95% CI 0.81–0.84) when compared with an individual who rejects vaccination. Lastly, the estimated probability that an individual delaying vaccination has higher trust as opposed to an individual refusing it is 0.68 (95% CI 0.65–0.71).

### 3.3. Alternative Medicine

As shown in Table 2, the proportion of COVID-19 vaccine acceptance was significantly (and greatly) lower among people who used alternative medicine as their main source of care, as compared with all the others respondents (64.4% vs. 86.3%) (Rao and Scott-corrected *χ*^2^ test = 145.56, *p*-value < 0.001). Consistently, seeing naturopaths, homeopaths, and chiropractors was associated with lower vaccine acceptance (Rao and Scott-corrected *χ*^2^ test = 34.25, *p*-value < 0.001).

Vaccine acceptance was also confirmed to be significantly lower among respondents using aromatherapy (64.4%), homeopathic products (69.2%), and herbal remedies (76.5%) compared with those not using such products or dietary supplements (87.5%).

### 3.4. Trust in the Current Government

As illustrated in Figure 3, trust in government followed a significantly different distribution according to vaccine uptake status (Kruskal–Wallis test = 883.98, *p*-value < 0.001), with significant differences resulting from all post hoc pairwise comparisons across groups (uptake vs. delay: Conover–Iman test = 15.25, *p*-value < 0.001; uptake vs. refusal: Conover–Iman test = 28.41, *p*-value < 0.001; delay vs. refusal: Conover–Iman test = 11.68, *p*-value < 0.001). Indeed, the respondents who exhibited minimal trust (i.e., 1 or 2 points on a scale from 1 to 10) accounted for 67.3% of “no-vaccinators”, 40.0% of “delayers”, and 16.5% of “vaccinators”.

More generally, we estimated a probability of 0.65 (95% CI 0.63–0.67) that a vaccinated individual has a higher trust in the government compared with an individual who puts off getting vaccinated, which increases to 0.81 (95% CI 0.79–0.82) when compared with an individual who rejects vaccination. Lastly, the estimated probability that an individual delaying vaccination has higher trust as opposed to an individual refusing it is 0.67 (95% CI 0.65–0.70).

### 3.5. Voting Intentions

There was evidence of different voting intentions in persons who received, put off, and refused COVID-19 vaccination (Rao and Scott-corrected *χ*^2^ test = 18.15, *p*-value < 0.001). In particular, as illustrated in Figure 4 and confirmed by post hoc adjusted residuals, intention to vote for right-wing parties was significantly lower among people accepting vaccination (24.7%) than among people delaying or refusing it (30.6% and 32.6%, respectively). Conversely, intention to vote for left-wing parties was significantly lower among people delaying or refusing vaccination (9.1% and 5.3%, respectively) than among people accepting it (23.8%). We also found a significantly lower intention to vote for Italy’s independent/anti-establishment party (*Movimento 5 Stelle*) in those opposing vaccination (8.0%) compared with those accepting or delaying it (13.0% and 14.0%, respectively). Lastly, abstentionism was significantly different in the three study groups (uptake: 12.0%; delay: 19.9%; refusal: 27.3%).

### 3.6. Policy Instruments

When the study participants were asked about their opinion on the policy instruments that the government should adopt to push the vaccination campaign (Table 3), those most in favor of introducing either mandatory vaccination or penalties were, as expected, the ones who got vaccinated as soon as possible (mandate: 43.6%; penalties: 8.2%), as compared with those who put off vaccination (mandate: 9.7%; penalties: 3.5%) and refused it (mandate: 2.8%; penalties: 1.6%). Conversely, “delayers” and “no-vaccinators” were more in favor of the principle that the government should simply make vaccines and information available, without influencing individual decisions (62.4% and 82.9%, respectively). However, “no-vaccinators” were also against the introduction of positive incentives (12.6%), while such an instrument was more appreciated by “delayers” (24.4%) and, to a lesser extent, by those accepting vaccination (18.5%).

### 3.7. Self-Declared Motivations

As shown in Table 4, among the 8315 individuals who were vaccinated against COVID-19 as soon as possible, 56.3% did so mainly to avoid severe illness, 19.0% to protect dear ones’, 11.8% from a sense of civic duty, 9.3% to return to normal life, and 3.5% for professional ethics. Among the 803 individuals who intentionally delayed vaccination, 55.0% did so mainly out of fear of side effects, 26.0% for obtaining more information about safety of COVID-19 vaccines, 8.9% for lack of time and/or difficulties accessing healthcare, 5.0% for opposition to vaccines in general, 3.4% because COVID-19 was not perceived as a serious disease, and 1.6% as a form of protest against government. Among the 673 individuals who refused to be vaccinated, 55.6% did so mainly because of fear of side effects, 27.2% due to lack of information about the safety of COVID-19 vaccines, 6.4% because COVID-19 was not perceived as a serious disease, 5.3% for opposition to vaccines in general, 3.4% for fear of needles and/or doctors, and 2.2% as a form of protest against government.

## 4. Discussion

The present work reports data from a survey conducted in the spring of 2022 on a large national sample of 10,000 adults from Italy. Although the survey contained questions pertaining to several vaccines, this article focuses on the section of the questionnaire pertaining to the COVID-19 vaccine, exploring vaccine refusal, uptake, and hesitancy and its possible drivers in the field of political or personal individual choices.

The pandemic and the availability of COVID-19 vaccines clearly reignited the debate and led to a further flowering of scientific publications on the topic of vaccine hesitancy. Indeed, compared with other vaccine policies, the SARS-CoV-2 immunization campaign added certain elements of complexity [27]: on the one hand, knowledge of the virus was limited, and there were gaps in understanding of the dynamics of its transmission; on the other hand, the need to respond to what was an emergency situation meant that the vaccines had been subjected to clinical trials and then approved rapidly. While several large trials carefully assessed safety and efficacy, there remained some uncertainties in the early post-licensure periods. These included impact on transmission, existence of rare serious adverse events, and their impact on COVID epidemiology in different sectors of the population [28]. These characteristics could reasonably be expected to increase vaccine hesitancy, not just among those who oppose vaccines in general, but also among individuals who are not anti-vaccine in principle but base their decisions on the available information and on the greater or lesser fear of adverse events following immunization.

This research shows that it makes sense to treat the “vaccine-hesitant” population not as one homogeneous category, but as an entity composed of several sub-categories, each showing skepticism toward vaccination for different reasons and with different beliefs and levels of confidence. In our analysis, we distinguished two (macro)categories of vaccine-hesitant people: “delayers” and “no-vaccinators”. Although these two categories shared some common traits (e.g., they are concentrated in the middle age groups and make extensive use of alternative medicine), we also found distinctiveness. “Delayers” were concentrated in the 25–44 age group, were more likely to have a postgraduate degree, showed moderate distrust of science and government, and were largely supportive of government incentives to promote vaccination. In contrast, “no-vaccinators” were predominantly female, had low educational attainment (less than a high school diploma), showed high distrust of science and government, and were unfavorable to the introduction of incentives for those who vaccinate. We have distinguished between two categories only, but future research may go further by segmenting in more detail those on the vaccine acceptance spectrum.

Previous studies established that attitudes toward COVID-19 vaccination can be influenced by sociodemographic variables, such as gender, age, and level of education both in Italy [29,30,31,32,33,34] and worldwide [35,36,37,38,39,40,41].

As some previous results underscore, hesitancy toward the COVID-19 vaccine is more pronounced in females than in men [42,43].

With regard to age, several surveys conducted around the world [36,38,44,45] show that in most countries the propensity to vaccinate against COVID-19 increases with age: older people are expected to be more willing to be vaccinated due to higher risks of severe infection [38], while vaccine uptake is significantly lower among the young [41]. However, some previous surveys conducted in Italy [32,46] paint a different picture compared with other countries. In Italy, the relationship between age groups and propensity to vaccinate against COVID-19 was described as a U-shape [33]: the most likely to vaccinate are those under 35 and over 60, while vaccine refusal is concentrated in the 35–50 age group. The data from our survey confirm the U-shaped relationship already found in Italy. It should be borne in mind when reading this result that in Italy, in early January 2022 a vaccination obligation was introduced for the entire over-50 population.

Previous studies focusing on the COVID-19 vaccine agree that in most countries, including Italy, vaccine hesitancy is significantly associated with lower educational attainment [30,32,38,40,41,47,48]. The findings of our survey paint a somewhat more nuanced picture thanks to our proposed distinction between “delayers” and “no-vaccinators”. Those with a lower educational qualification (less than high school diploma) were more likely to be “no-vaccinators”; those with a university degree (bachelor or master’s degree) were more likely to be vaccinators; those with a qualification beyond master’s degree were less likely to be vaccinators and more likely to be “delayers”. Indeed, it is to be expected that the category of “delayers” is largely made up of individuals who postpone vaccination for one of the following reasons: because they face practical impediments or have low motivation; because of a rational calculation of costs and benefits; and because of free-riding.

Several contributions on vaccine hesitancy have also shown its association with low levels of trust in science [38,44,45,48]. All surveys conducted so far in Italy regarding those opposing vaccination agree on one aspect: those who have higher trust in science are more willing to get vaccinated [29,32,34,46]. This correlation is found in many other countries, with some studies identifying trust in science as the strongest predictor of vaccine acceptance [32,36,38,45]. In this respect, our survey fully confirms the initial expectations: vaccinated individuals had higher values of trust in science; “no-vaccinators” were more likely to have low trust in science; “delayers” were in the middle.

Some previous research surveys conducted worldwide suggest an association between vaccine hesitancy and habitual recourse to complementary and alternative medicine [8,49,50,51,52]. Because, to the best of our knowledge, no scientific survey conducted so far has explored this relationship in Italy, a battery of questions was included in the questionnaire to investigate the respondents’ propensity to resort to natural treatments and alternative medicine (i.e., homeopathy, naturopathy, chiropractic, or osteopathy). Respondents to our questionnaire were thus asked if they used alternative medicine “as their main source of care”. The results are clear, and the correlation is statistically significant: those who use alternative medicine were more likely to be “delayers” or “no-vaccinators”. In short, the Italian case seems to confirm what has already been shown in other countries.

An added value of our survey is that it included some questions on what we might call “political orientation”. A first question concerned trust in the national government in charge. At the time the survey was administered (the spring of 2022), the Draghi government (a coalition government defined as one of “national unity” and supported by all major parties except the right-wing party of *Fratelli d’Italia*) was in office.

The results of this survey confirm that in Italy, as well as in other countries, certain “political variables” are intertwined with attitudes toward vaccination [32,44,45,48]. Trust toward the government was low among “delayers”, and even lower among “no-vaccinators”.

A second political variable concerned voting intentions and party identification. The question was formulated as follows: “If there were to be political elections tomorrow, which party would you vote for”? Previous surveys conducted in Italy investigated the link between vaccination attitudes and political orientation, asking respondents for generic placement on the right–left axis [41,52,53]. In this survey, respondents were asked to indicate a party specifically and were also given the option to define themselves as “not voting”, since the area of abstentionism is acquiring an increasingly relevant role in Italian politics [54].

Voters of *Fratelli d’Italia* (a far-right party) and *Movimento 5 Stelle* (an independent/anti-establishment party) were over-represented among “delayers”. “No-vaccinators” would have voted more frequently for *Fratelli d’Italia* or other right-wing parties that were not in government when the survey was carried out, and those who had no intention to vote were more likely to be “delayers” and even more to be “no-vaccinators”. It can therefore be argued that the willingness to vaccinate is conditioned by the individuals’ attitude toward authority, institutions, and democratic rules of play. In other words, choosing not to get vaccinated can take on a political, protesting value. However, turning to the dimension of policy instruments (more details on this in the next paragraph), although “delayers” and even more “no-vaccinators” were against both mandatory vaccination and penalties (for those who do not vaccinate), “no-vaccinators” were less sensitive than “delayers” to positive incentives (for those who vaccinate). This finding can be a good starting point to advance some policy recommendations for public decision makers.

As previously mentioned, a third political variable concerned the opinion toward the “policy instruments” that the government can use to promote the vaccination campaign. Indeed, policy makers can choose from a variety of policy instruments, which can be placed on a “ladder of intrusiveness” [55,56,57], starting from voluntary tools (simple information and persuasion), through material incentives and disincentives, to highly coercive tools, such as the introduction of the vaccination mandate. An emerging strand in the literature is interested in understanding—depending on the specific vaccine and social context—the effects of different instruments on vaccination behaviors [3,58,59]. With the intention of contributing to this debate, the statements on governance and instruments that should be adopted in health emergencies with which the participants most agreed were analyzed.

As might have been expected, those most in favor of introducing mandatory vaccination or penalties for the unvaccinated were “vaccinators”; such measures were decidedly less supported by “delayers” and even less by “no-vaccinators”. Instead, the latter two categories were more in favor of the principle that the government should simply make vaccines and information available, without influencing individual decisions. However, “no-vaccinators” were also against the introduction of positive incentives, whilst such an instrument was appreciated by “delayers”. Hence, it is among delayers that we are most likely to find a higher concentration of both “fence-sitters” and “calculating” individuals since—as mentioned earlier—both of these profiles are more sensitive to the introduction of incentives.

Figures from our survey could help further understand why people refuse vaccinations and what people who are hesitant (“delayers” in our survey) think. The majority of people who declared that they had been vaccinated as soon as possible, not only gave reasons related to the evaluation of the benefits, but also showed a strong sense of collective responsibility. Among “delayers”, motivations related to calculation prevailed. Constraints included lack of trust toward vaccines in general, hostility toward the government and, residually, complacency. A similar profile characterized the “no-vaccinators”, but with double the percentages for reasons related to complacency and hostility toward the government.

### Strengths and Limitations

In this research, we distinguished the study sample into three categories: individuals who were in favor of the vaccine (and vaccinated as soon as possible), individuals who intentionally postponed vaccination (whom we called “delayers”), and those who—more than a year after the start of the vaccination campaign—had not been vaccinated (whom we called “no-vaccinators”). As shown before, most surveys conducted so far simply distinguish between two categories (individuals in favor versus against the vaccine). Therefore, the distinction into three categories represents the first added value of the present analysis. This tripartite classification makes it possible to grasp the different gradations of vaccine aversion and explore the various determinants that fuel reluctance to vaccinate.

With respect to what we already know about the Italian context, it should be noted that most of the surveys conducted so far regarding COVID-19 vaccines have used non-representative samples [29,32,34,60,61,62,63,64]. Only a few published studies used a nationally representative sample [46], while some others used regionally representative samples [33,65]. None of them, however, were based on a number of respondents as large as ours.

Last but not least, previous studies in countries other than Italy found an association between political orientation and attitudes toward vaccination [36,37,38,66,67,68,69,70]. However, the relationship between “political variables” and vaccination behavior is still little explored—with a few exceptions [41,53]—in Italy. For this reason, our survey investigated the opinions of respondents with reference to both politics (the degree of trust in the current government; the political party with which they identified) and policy (which policy instruments were considered most appropriate to promote vaccination). Having included these “political variables” in the analysis is a further element of originality for the present research.

We acknowledge that this work also has some limitations. First, being a cross-sectional study, it only provides statistical associations between variables and does not enable causal inferences. Second, despite the sample being chosen to be representative of Italy’s demographics, this was an online paid survey that may have attracted people to receive some extra income, leading to a possible over-representation of lower socioeconomic classes. Third, we did not investigate household income, religion, and other sensitive social characteristics that may have jeopardized the size, power, and representativeness of our sample.

## 5. Conclusions

This article about political beliefs and attitudes toward COVID-19 vaccines contains findings and suggestions characterized by a broad scope that can be extended to other vaccination campaigns.

For a vaccination campaign to be successful, it is essential to segment the target population according to different orientations toward vaccination [71]. Since acceptance and refusal of vaccines are highly context-dependent [8] and each vaccine is unique, it is necessary to estimate on a case-by-case basis the proportions of the different profiles of people accepting, refusing, or delaying vaccinations. Only detailed population mapping can in fact enable policy makers to select the most effective and appropriate policy instruments. It is plausible that each category of “hesitant” individuals—depending on the reasons they defer or reject the vaccine—is sensitive toward some policy instruments but insensitive toward others. An economic incentive, just to give an example, will be motivating for “calculator” individuals, but not for those who oppose the vaccine on ideological grounds or to express distrust of the current government. This implies that some instruments may be found to be costly economically and/or politically but at the same time poorly effective overall. This is something that policy makers should consider when shaping vaccine strategies and choosing policy instruments.

Further research on this topic should employ machine learning algorithms for classification, such as random forests and support vector networks, in order to bypass parametric assumptions for multiple regression and identify clusters of subjects sharing some common features who are likely to either receive, delay, or refuse vaccination.

## Figures and Tables

**Figure 1 vaccines-11-00839-f001:**
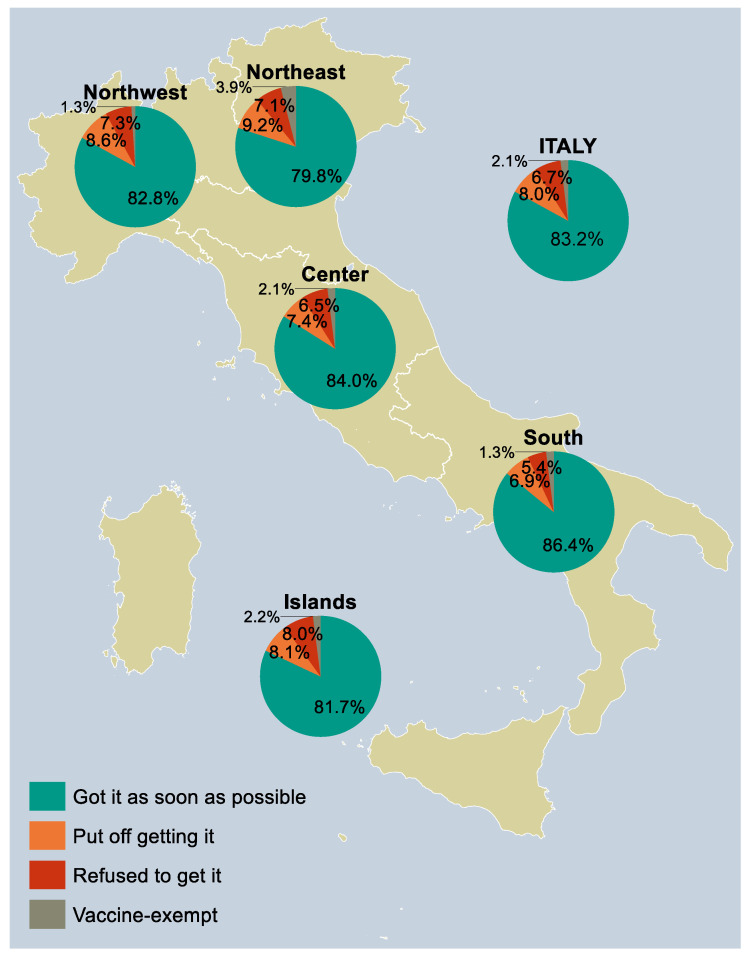
COVID-19 vaccine uptake in the study sample between January 2021 and April/May 2022 (*n* = 10,000), overall and by NUTS statistical region. Percentages were post-stratified by using the distribution of Italy’s resident adult population by NUTS region, gender, and age group as of 1 January 2022. *Notes:* Northwestern Italy includes Piedmont, Aosta Valley, Lombardy, and Liguria; Northeastern Italy includes Trentino-South Tyrol, Veneto, Friuli-Venezia Giulia, and Emilia-Romagna; Central Italy includes Tuscany, Umbria, Marche, and Lazio; Southern Italy includes Abruzzo, Molise, Campania, Apulia, Basilicata, and Calabria; Insular Italy includes Sicily and Sardinia. *Abbreviations:* COVID-19, coronavirus disease 2019; NUTS, Nomenclature of Territorial Units for Statistics.

**Figure 2 vaccines-11-00839-f002:**
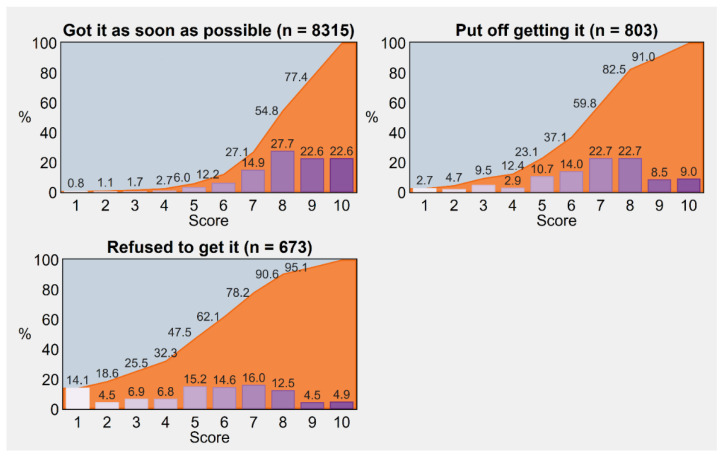
Trust in science scored on a scale from a minimum of 1 to a maximum of 10, by COVID-19 vaccine uptake status. Percentages were post-stratified by using the distribution of Italy’s resident adult population by NUTS region, gender, and age group as of 1 January 2022. *Notes:* Cumulative frequencies are displayed with the aid of area charts in the background. Trust among vaccine-exempt individuals is not displayed due to small numbers (*n* = 209). *Abbreviations:* COVID-19, coronavirus disease 2019; NUTS, Nomenclature of Territorial Units for Statistics.

**Figure 3 vaccines-11-00839-f003:**
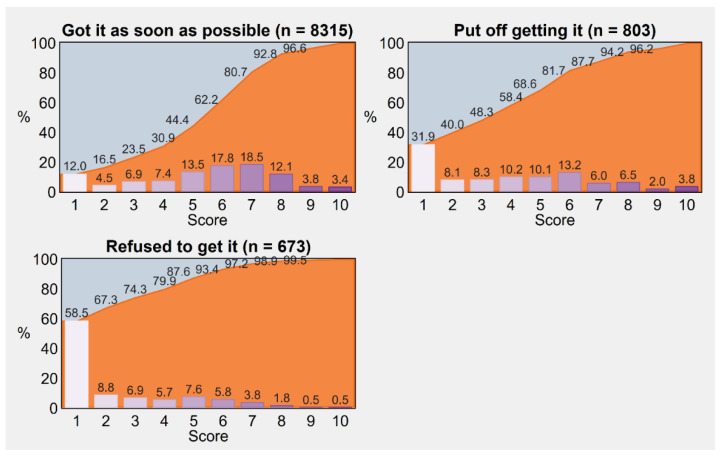
Trust in the Italian government scored on a scale from a minimum of 1 to a maximum of 10, by COVID-19 vaccine uptake status. Percentages were post-stratified by using the distribution of Italy’s resident adult population by NUTS region, gender, and age group as of 1 January 2022. *Notes:* Cumulative frequencies are displayed with the aid of area charts in the background. Trust among vaccine-exempt individuals is not displayed due to small numbers (*n* = 209). *Abbreviations:* COVID-19, coronavirus disease 2019; NUTS, Nomenclature of Territorial Units for Statistics.

**Figure 4 vaccines-11-00839-f004:**
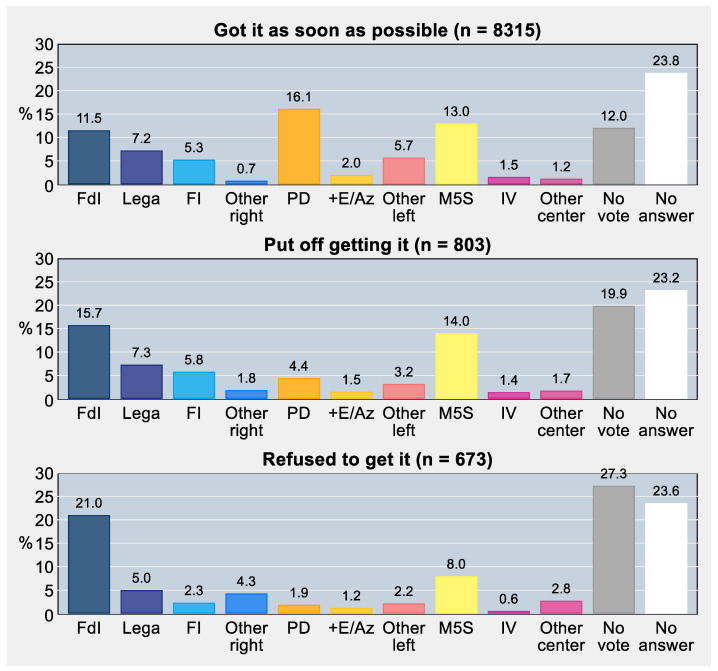
Voting intention as of April/May 2022 by COVID-19 vaccine uptake status. Percentages were post-stratified by using the distribution of Italy’s resident adult population by NUTS region, gender, and age group as of 1 January 2022. *Notes:* Voting intention among vaccine-exempt individuals is not displayed due to small numbers (*n* = 209); *FdI*, *Fratelli d’Italia* (Brothers of Italy); *FI*, *Forza Italia* (Forward Italy); *PD*, *Partito Democratico* (Democratic Party); *+E*, *Più Europa* (More Europe); *Az*, *Azione* (Action); *M5S*, *Movimento 5 Stelle* (Five Star Movement); *IV*, *Italia Viva* (Italy Alive). *Abbreviations:* COVID-19, coronavirus disease 2019; NUTS, Nomenclature of Territorial Units for Statistics.

**Table 1 vaccines-11-00839-t001:** Sociodemographic characteristics of the study sample, overall and by COVID-19 vaccine uptake status. Counts and percentages were obtained by weighting all observed values with post-stratification weights based on the distribution of Italy’s resident adult population by NUTS region, gender, and age group.

Characteristic	All (*n* = 10,000)	Vaccinated as Soon as Possible (*n* = 8315)	Put Off Getting Vaccinated (*n* = 803)	Refused to Get Vaccinated (*n* = 673)	Vaccine-Exempt (*n* = 209)	*p*-Value
Gender						<0.001
Male	4823 (100.0%)	4051 (84.0%)	356 (7.4%)	281 (5.8%) ^(−)^	135 (2.8%) ^(+)^	
Female	5158 (100.0%)	4251 (82.4%)	445 (8.6%)	389 (7.5%) ^(+)^	73 (1.4%) ^(−)^	
Non-binary	13 (100.0%)	9 (69.2%)	1 (7.7%)	3 (23.1%)	0 (0.0%)	
Prefer not to say	6 (100.0%)	4 (66.7%)	1 (16.7%)	0 (0.0%)	1 (16.7%)	
Age group, y						<0.001
18–24	824 (100.0%)	709 (86.0%)	61 (7.4%)	31 (3.8%) ^(−)^	23 (2.8%)	
25–34	1254 (100.0%)	997 (79.5%) ^(−)^	155 (12.4%) ^(+)^	70 (5.6%)	32 (2.6%)	
35–44	1461 (100.0%)	1125 (77.0%) ^(−)^	164 (11.2%) ^(+)^	127 (8.7%)	45 (3.1%)	
45–54	1884 (100.0%)	1527 (81.1%)	166 (8.8%)	157 (8.3%)	34 (1.8%)	
55–64	1756 (100.0%)	1503 (85.6%)	135 (7.7%)	99 (5.6%)	19 (1.1%)	
≥65	2821 (100.0%)	2454 (87.0%) ^(+)^	122 (4.3%) ^(−)^	189 (6.7%)	56 (2.0%)	
Place of residence degree of urbanization						<0.001
City (densely populated area)	3910 (100.0%)	3317 (84.8%) ^(+)^	291 (7.4%)	243 (6.2%)	59 (1.5%)	
Town or suburb (intermediate density area)	4687 (100.0%)	3931 (83.9%)	354 (7.6%)	331 (7.1%)	71 (1.5%) ^(−)^	
Rural area (thinly populated area)	1403 (100.0%)	1067 (76.1%) ^(−)^	158 (11.3%) ^(+)^	99 (7.1%)	79 (5.6%) ^(+)^	
Educational attainment						<0.001
Less than high school diploma	1487 (100.0%)	1218 (81.9%)	119 (8.0%)	139 (9.3%) ^(+)^	11 (0.7%) ^(−)^	
High school diploma	5742 (100.0%)	4794 (83.5%)	458 (8.0%)	404 (7.0%)	86 (1.5%) ^(−)^	
Academic degree	1987 (100.0%)	1722 (86.7%) ^(+)^	137 (6.9%)	93 (4.7%) ^(−)^	35 (1.8%)	
Post-graduate/Doctorate degree	784 (100.0%)	581 (74.1%) ^(−)^	89 (11.4%) ^(+)^	37 (4.7%)	77 (9.8%) ^(+)^	

^(+)^ Adjusted residual ≥3.29 (≥*z*_0.001/2_), that is, frequency is significantly greater than what would be expected if the null hypothesis of independence was true. ^(−)^ Adjusted residual ≤3.29 (≤*z*_0.001/2_), that is, frequency is significantly lower than what would be expected if the null hypothesis of independence was true. *Abbreviations:* COVID-19, coronavirus disease 2019; NUTS, Nomenclature of Territorial Units for Statistics.

**Table 2 vaccines-11-00839-t002:** Use of alternative medicine in the study sample, overall and by COVID-19 vaccine uptake status. Counts and percentages were obtained by weighting all observed values with post-stratification weights based on the distribution of Italy’s resident adult population by NUTS region, gender, and age group.

Characteristic	All (*n* = 10,000)	Vaccinated as Soon as Possible (*n* = 8315)	Put Off Getting Vaccinated (*n* = 803)	Refused to Get Vaccinated (*n* = 673)	Vaccine-Exempt (*n* = 209)	*p*-Value
Use of homeopathy, naturopathy, chiropractic or osteopathy as the main source of care						<0.001
Yes	1439 (100.0%)	927 (64.4%) ^(−)^	217 (15.1%) ^(+)^	187 (13.0%) ^(+)^	108 (7.5%) ^(+)^	
No	8561 (100.0%)	7388 (86.3%) ^(+)^	586 (6.8%) ^(−)^	486 (5.7%) ^(−)^	101 (1.2%) ^(−)^	
Visit to one of the following practitioners over the last year						<0.001
Osteopath	773 (100.0%)	636 (82.3%)	81 (10.5%)	35 (4.5%)	21 (2.7%)	
Homeopath	388 (100.0%)	251 (64.7%) ^(−)^	61 (15.7%) ^(+)^	38 (9.8%)	38 (9.8%) ^(+)^	
Naturopath	233 (100.0%)	135 (57.9%) ^(−)^	46 (19.7%) ^(+)^	29 (12.4%) ^(+)^	23 (9.9%) ^(+)^	
Chiropractor	219 (100.0%)	153 (69.9%) ^(−)^	27 (12.3%)	14 (6.4%)	25 (11.4%) ^(+)^	
None of the above	8387 (100.0%)	7140 (85.1%) ^(+)^	588 (7.0%) ^(−)^	557 (6.6%)	102 (1.2%) ^(−)^	
Use of one of the following products over the last year						<0.001
Supplements (e.g., proteins, vitamins, minerals)	5300 (100.0%)	4419 (83.4%)	410 (7.7%)	387 (7.3%)	84 (1.6%) ^(−)^	
Herbal remedies	757 (100.0%)	579 (76.5%) ^(−)^	90 (11.9%) ^(+)^	57 (7.5%)	31 (4.1%) ^(+)^	
Homeopathic products	419 (100.0%)	290 (69.2%) ^(−)^	63 (15.0%) ^(+)^	33 (7.9%)	33 (7.9%) ^(+)^	
Aromatherapy (e.g., essential oils)	239 (100.0%)	154 (64.4%) ^(−)^	38 (15.9%) ^(+)^	19 (7.9%)	28 (11.7%) ^(+)^	
None of the above	3285 (100.0%)	2873 (87.5%) ^(+)^	202 (6.1%) ^(−)^	177 (5.4%) ^(−)^	33 (1.0%) ^(−)^	

^(+)^ Adjusted residual ≥3.29 (≥*z*_0.001/2_), that is, frequency is significantly greater than what would be expected if the null hypothesis of independence was true. ^(−)^ Adjusted residual ≤3.29 (≤*z*_0.001/2_), that is, frequency is significantly lower than what would be expected if the null hypothesis of independence was true. *Abbreviations:* COVID-19, coronavirus disease 2019; NUTS, Nomenclature of Territorial Units for Statistics.

**Table 3 vaccines-11-00839-t003:** Opinion on vaccination policies in the study sample, overall and COVID-19 vaccine uptake status. Counts and percentages were obtained by weighting all observed values with post-stratification weights based on the distribution of Italy’s resident adult population by NUTS region, gender, and age group.

Answer	All (*n* = 10,000)	Vaccinated as Soon as Possible (*n* = 8315)	Put Off Getting Vaccinated (*n* = 803)	Refused to Get Vaccinated (*n* = 673)	Vaccine-Exempt (*n* = 209)
Governments must make vaccines mandatory during health emergencies	3733 (37.3%)	3626 (43.6%) ^(+)^	78 (9.7%) ^(−)^	19 (2.8%) ^(−)^	10 (4.8%) ^(−)^
Governments should restrict themselves to making vaccines available and providing information, without influencing individual decisions	3659 (36.6%)	2471 (29.7%) ^(−)^	501 (62.4%) ^(+)^	558 (82.9%) ^(+)^	129 (61.7%) ^(+)^
Governments must offer incentives to those who get vaccinated, but without imposing any vaccine mandate	1878 (18.8%)	1539 (18.5%)	196 (24.4%)	85 (12.6%) ^(−)^	58 (27.8%) ^(+)^
Governments must use penalties for those who decide not to get vaccinated	730 (7.3%)	679 (8.2%) ^(+)^	28 (3.5%) ^(−)^	11 (1.6%) ^(−)^	12 (5.7%)

^(+)^ Adjusted residual ≥3.29 (≥*z*_0.001/2_), that is, frequency is significantly greater than what would be expected if the null hypothesis of independence was true. ^(−)^ Adjusted residual ≤3.29 (≤*z*_0.001/2_), that is, frequency is significantly lower than what would be expected if the null hypothesis of independence was true. *Notes:* Rao and Scott-corrected *χ*^2^ test of independence between opinion and uptake is equal to 138.97 (*p*-value < 0.001). *Abbreviations:* COVID-19, coronavirus disease 2019; NUTS, Nomenclature of Territorial Units for Statistics.

**Table 4 vaccines-11-00839-t004:** Reasons for receiving COVID-19 vaccination as soon as possible, putting off getting vaccinated, and refusing vaccination. Counts and percentages were obtained by weighting all observed values with post-stratification weights based on the distribution of Italy’s resident adult population by NUTS region, gender, and age group.

Answer	All	Importance Ranking			
		1st	2nd	3rd	>3rd
Vaccinated as soon as possible					
Avoiding severe illness	8315 (100.0%)	4681 (56.3%)	929 (11.2%)	688 (8.3%)	2017 (24.3%)
Protecting dear ones	8315 (100.0%)	1583 (19.0%)	3095 (37.2%)	1087 (13.1%)	2550 (30.7%)
Sense of civic duty	8315 (100.0%)	984 (11.8%)	1371 (16.5%)	2045 (24.6%)	3915 (47.1%)
Returning to normal life	8315 (100.0%)	773 (9.3%)	770 (9.3%)	1730 (20.8%)	5042 (60.6%)
Professional ethics	8315 (100.0%)	294 (3.5%)	313 (3.8%)	471 (5.7%)	7237 (87.0%)
Put off vaccination					
Fear of side effects	803 (100.0%)	442 (55.0%)	96 (12.0%)	37 (4.6%)	228 (28.4%)
Obtaining more info about safety	803 (100.0%)	209 (26.0%)	236 (29.4%)	49 (6.1%)	309 (38.5%)
Lack of time and/or difficulties of access	803 (100.0%)	72 (8.9%)	36 (4.5%)	64 (8.0%)	631 (78.6%)
Opposition to vaccines in general	803 (100.0%)	40 (5.0%)	73 (9.1%)	67 (8.3%)	623 (77.6%)
COVID-19 not a serious disease	803 (100.0%)	27 (3.4%)	43 (5.4%)	75 (9.3%)	658 (81.9%)
Protest against government	803 (100.0%)	13 (1.6%)	21 (2.6%)	45 (5.6%)	724 (90.2%)
Refused vaccination					
Fear of side effects	673 (100.0%)	374 (55.6%)	111 (16.5%)	45 (6.7%)	143 (21.2%)
Lack of information about safety	673 (100.0%)	183 (27.2%)	261 (38.8%)	49 (7.3%)	180 (26.7%)
COVID-19 not a serious disease	673 (100.0%)	43 (6.4%)	26 (3.9%)	90 (13.4%)	514 (76.4%)
Opposition to vaccines in general	673 (100.0%)	36 (5.3%)	27 (4.0%)	66 (9.8%)	544 (80.8%)
Fear of needles and/or doctors	673 (100.0%)	23 (3.4%)	20 (3.0%)	14 (2.1%)	616 (91.5%)
Protest against government	673 (100.0%)	15 (2.2%)	26 (3.9%)	60 (8.9%)	572 (85.0%)

*Notes:* Respondents were asked to pick three reasons in order of importance. *Abbreviations:* COVID-19, coronavirus disease 2019; NUTS, Nomenclature of Territorial Units for Statistics.

## Data Availability

All data is provided in the manuscript.

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
