# Peer review of "COVID-19 Vaccine Refusal and Delay among Adults in Italy: Evidence from the OBVIOUS Project, a National Survey in Italy"

_vaccines, 2023, doi:10.3390/vaccines11040839_

Round 1

Reviewer 1 Report

See file attached

Reviewer 2 Report

Even if this paper is based on a cross-sectional, non-representative sample and online paid survey, I found it extremely informative, well written, with an open-minded discussion.

This paper is excellent!

Author Response

Response: Thank you for your kind and enthusiastic comment. We are glad to know you appreciated our work.

Reviewer 3 Report

There was one surprising finding – for me – here, namely that women were more hesitant to get vaccinated. I recalled that women were more accepting of vaccines in Hungary. A quick search resulted in ambiguous results. Two published surveys are in alignment with the finding reported here: man accepts vaccines more readily.

https://www.nature.com/articles/s41598-022-26824-5

https://link.springer.com/article/10.1186/s12889-021-12386-0

But a representative survey in Hungary showed that women were less hesitant to take the vaccine (https://link.springer.com/article/10.1007/s10865-022-00314-5). And actual vaccination data also show that more women are vaccinated than men (https://www.mdpi.com/2076-393X/10/7/1009).

On politics you might find the following article interesting: https://www.mdpi.com/2076-393X/10/5/789

(I was not involved with any of the above publications)

I would avoid abbreviations as much as possible. Vaccine hesitation does not need to be abbreviated as VH.

Ref19: It now has volume and pages. 48(1):24-29

Round 2

Reviewer 1 Report

The authors have addressed all my comments satisfactorily. The article can be accepted for publication in its present form.